# Validity of Spatio-Temporal Gait Parameters in Healthy Young Adults Using a Motion-Sensor-Based Gait Analysis System (ORPHE ANALYTICS) during Walking and Running

**DOI:** 10.3390/s23010331

**Published:** 2022-12-28

**Authors:** Yuki Uno, Issei Ogasawara, Shoji Konda, Natsuki Yoshida, Naoki Otsuka, Yuya Kikukawa, Akira Tsujii, Ken Nakata

**Affiliations:** 1Department of Health and Sport Sciences, Graduate School of Medicine, Osaka University, Suita 565-0871, Osaka, Japan; 2ORPHE Inc., Shibuya 151-0053, Tokyo, Japan; 3Department of Sports Medical Biomechanics, Graduate School of Medicine, Osaka University, Suita 565-0871, Osaka, Japan

**Keywords:** validation study, reliability, wearable devices, inertial measurement unit (IMU), foot-mounted sensor, foot-strike pattern

## Abstract

Motion sensors are widely used for gait analysis. The validity of commercial gait analysis systems is of great interest because calculating position/angle-level gait parameters potentially produces an error in the integration process of the motion sensor data; moreover, the validity of ORPHE ANALYTICS, a motion-sensor-based gait analysis system, has not yet been examined. We examined the validity of the gait parameters calculated using ORPHE ANALYTICS relative to those calculated using conventional optical motion capture. Nine young adults performed gait tasks on a treadmill at speeds of 2–12 km/h. The three-dimensional position data and acceleration and angular velocity data of the feet were collected. The gait parameters were calculated from motion sensor data using ORPHE ANALYTICS, and optical motion capture data. Intraclass correlation coefficients [ICC_(2,1)_] were calculated for relative validities. Eight items, namely, stride duration, stride length, stride frequency, stride speed, vertical height, stance phase duration, swing phase duration, and sagittal angle_IC_ exhibited excellent relative validities [ICC_(2,1)_ > 0.9]. In contrast, sagittal angle_TO_ and frontal angle_IC_ demonstrated good [ICC_(2,1)_ = 0.892–0.833] and moderate relative validity [ICC_(2,1)_ = 0.566–0.627], respectively. ORPHE ANALYTICS was found to exhibit excellent relative validities for most gait parameters. These results suggest its feasibility for gait analysis outside the laboratory setting.

## 1. Introduction

Motion sensors (inertia sensors) have become widely used for the motion measurement of any type of object owing to their rapid technological development and have recently been applied to gait analysis [1]. Motion sensors are highly compact and lightweight, allowing their attachment to the body without spatially restricting the object of measurement. These features potentially overcome the significant limitations of optical cameras and reflective marker-based motion capture systems commonly used in motion measurement, which are only feasible in a limited laboratory space [2]. The motion sensor will expand the range of applications, such as a more natural gait assessment in daily life and frequent measurements without visiting a specific facility [3].

Several commercial packages that take advantage of motion-sensor compactness and lightweight are currently available [4,5,6,7]. One commercial product is ORPHE ANALYTICS (ORPHE Inc., Tokyo, Japan), which is a gait analysis system that can evaluate position/angle-level gait parameters (such as stride length and foot-ground angle) using the acceleration and angular velocity data of shoes during walking and running. These data are collected using a 20-g sensor mounted on a shoe’s midsole or instep. Therefore, provided individuals wear their smart shoes, assessing their gait patterns without constraining the measurement environment is possible.

Position/angle-level gait parameters are commonly utilized in conventional gait assessments [8] because conventional gait analysis has frequently used optical motion capture, which can measure the position and angle of the targeted body segment or shoe. Furthermore, position/angle-level gait parameters are easier to understand than derivative quantities such as accelerations and angular velocities. This background probably motivates the calculation of position/angle-level gait parameters in motion-sensor-based gait analysis. However, because motion sensors cannot directly measure the position/angle-level kinematic properties, numerical integration of the measured acceleration or angular velocity data is necessary. The numerical integration process poses certain challenges, such as the production of numerical errors. These errors are affected by sensor specifications [9,10] and mounting position [11,12]. Therefore, the use of motion-sensor-based commercial gait analysis systems to determine how consistently position/angle-level gait parameters are obtained from motion sensors is of great interest. ORPHE ANALYTICS has not yet been used to verify how consistent the position/angle-level gait parameters calculated using this motion-sensor-based system and its software are with the corresponding parameters obtained using the conventional optical motion capture.

Therefore, this study aimed to validate ORPHE ANALYTICS by assessing the agreement of position/angle-level gait parameters between ORPHE ANALYTICS and optical motion capture during walking and running in healthy participants. We evaluated the relative validity between the gait parameters from the two different modalities using intraclass correlation coefficients (ICCs) and visualized them using Bland–Altman plots. We hypothesized that the gait parameters calculated using ORPHE ANALYTICS exhibit excellent relative validities (ICC > 0.9) to those calculated using optical motion capture. 

## 2. Materials and Methods

### 2.1. Participants

Nine healthy volunteers who had not undergone surgical treatment of the lower extremity within one year before the experiment participated in the study (six men and three women; mean age: 25.4 ± 2.2 y; mean height: 166.6 ± 9.7 cm; mean weight: 60.3 ± 10.7 kg; mean shoe size: 25.7 ± 1.3 cm). The Osaka University Hospital Ethics Committee approved this study (approval no. 19537), and all participants provided informed consent. 

### 2.2. Procedure

The participants wore shoes (SHIBUYA 2.0, ORPHE Inc., Tokyo, Japan) with midsole space to embed the motion sensor (ORPHE Inc., Tokyo, Japan; size: 45 mm×29 mm×14 mm; weight: 20 g; Figure 1a). This motion sensor samples the three-axial accelerations and angular velocities using a built-in sensor (LSM6DSOX, STMicroelectronics; acceleration: ±16 G, angular velocity: ±2000 °/s) at a sampling frequency of 200 Hz. The recorded data were stored in built-in flash memory. Two motion sensors were utilized for each foot. One sensor was embedded in the midsole space (plantar-embedded, Figure 1a), and the other was securely fixed onto the shoe instep using dedicated mounting equipment (instep-mounted, Figure 1b). Fourteen reflective markers were attached to the shoe landmarks to specify the posture and position of the motion sensor on the foot (Figure 1c).

Participants performed walking and running tasks on a treadmill (MyRun, Technogym, Cesena, Italy). The walking task required double support phases and was performed at target speeds of 2, 4, and 6 km/h. The running task required no double support phases and was performed at target speeds of 6, 9, and 12 km/h. The three foot-strike conditions, that is, the self-selected foot, forefoot, and rearfoot strike patterns, were performed only in the 12-km/h running task. The different foot-strike patterns’ purpose was to replicate natural running variability and increase the associated data variation. Foot-strike pattern differences have been reported to affect gait-event detection timing in gait analysis using foot-mounted motion sensors [13]. Two experimenters (Yuki Uno and Natsuki Yoshida) monitored the task requirements. At the beginning of each trial, the participant maintained a static standing position on the treadmill for 10 s and subsequently performed a vertical jump to synchronize the motion sensors and motion capture system. Thereafter, the participants were asked to increase the treadmill speed by themselves to the target speed and maintain the assigned gait speed for 1 min. Tasks were performed from the slower to the faster condition for the participants’ safety. Three-minute resting periods were provided between each trial to reduce the effect of fatigue. The participants could discontinue their trial at any time when they experienced difficulty completing further trials (see Appendix A for details of the tasks completed by the participants). The reflective marker positions were measured using 12 optical cameras (OptiTrack Prime 17 W, NaturalPoint, Inc., Corvallis, OR, USA) at a 360-Hz sampling frequency. Since we set the treadmill’s horizontal surface as the horizontal plane of the motion capture coordinate system, the vertical height of the treadmill’s horizontal surface was 0 m in the motion capture coordinate system. The three-axial accelerations and angular velocities of the four motion sensors (plantar/instep on both sides) were sampled at 200 Hz.

### 2.3. Data Processing

The marker position data were smoothed using a second-order Butterworth digital filter (low-pass, zero-lag, cut-off frequency: 5 Hz [14]). The motion sensor center position, anteroposterior axis, and mediolateral axis used in parameter calculation were defined as follows: the motion sensor center was the midpoint between the ALS and PMS markers. The anteroposterior axis was defined as the unit vector extending from the PLS marker to the ALS marker. The support vector was defined as the unit vector extending from the PMS to the PLS for the right foot and from the PLS to the PMS for the left foot. The vertical axis was defined as the cross-product of the support vector and anteroposterior axis vector. The mediolateral axis was defined as the cross-product of the anteroposterior and vertical axis vectors.

The timing of the initial foot contact (IC) for each gait cycle was defined as the instance at which the target marker’s peak vertical acceleration appeared [15]. The target marker was selected in every step according to the foot’s orientation when the vertical distance between any of the foot markers and the treadmill surface was <50 mm. The HEEL marker was assigned as the target marker when the angle between the motion sensor’s anteroposterior axis and global horizontal plane was <–15° (toe-up), the LMP marker was assigned when the angle ranged from −15° to −5° (near-flat), and the TOE marker was assigned when the angle was >−5° (toe-down) (Appendix A). The timing of the toe-off (TO) was defined when the TOE marker exceeded 10 mm above the minimum vertical height for each cycle [14]. One gait cycle was defined from one IC to the next for each foot [16]. The gait parameters listed in Table 1 were calculated using each gait cycle’s motion capture data.

The gait parameters based on the motion sensor data were calculated using ORPHE ANALYTICS (version 1.4.2; ORPHE Inc., Tokyo, Japan) for each gait cycle (Table 1).

We defined the gait cycles to be analyzed and calculated the gait-cycle detection ratio because ORPHE ANALYTICS did not detect the gait cycle ideally. The motion sensor and optical motion capture data were time-synchronized with the initial vertical jump’s landing timing, as described in the method. This procedure was not exact-time synchronization but was sufficient for gait-cycle-wise correspondence between modalities. To analyze the period after the treadmill speed reached the target speed, the first 15 gait cycles detected from the motion sensor data were discarded. The period from 350 ms before the 16th gait cycle to 350 ms after the last gait cycle was defined as the analysis period. Regarding the gait cycles calculated from the optical motion capture data, the gait cycles covered in the analysis period were analyzed. The gait-cycle detection ratio was calculated as the number of gait cycles for analysis by the motion sensor divided by that for analysis by optical motion capture.

Outlier processing was performed to exclude gait cycles containing abnormal gait-parameter values after calculating the gait-cycle detection ratio. Quartiles were calculated for each trial based on the stride length and duration obtained from the motion sensor and motion capture data. Gait cycles that included stride length or duration outside 1.5 times the interquartile range above the upper quartile point or below the lower quartile point were excluded from the analysis. The percentage of excluded gait cycles was subsequently calculated.

### 2.4. Statistical Analysis

The average gait-parameter values for each trial were calculated separately for the left and right sides, and statistical processing was performed. To evaluate the relative validity of the average gait-parameter values calculated from motion sensor data against those calculated from the motion capture, ICC_(2,1)_ values were calculated for the overall (walking and running), walking, and running conditions using average gait-parameter values. ICC_(2,1)_ values were interpreted as follows: excellent (> 0.90), good (0.75–0.90), moderate (0.50–0.75), and poor (< 0.50) [17].

To visualize the bias and precision between the gait parameters from the motion sensor data and those from the motion capture data, Bland–Altman plots [18] and histograms were drawn. The normality of differences in gait parameters from motion sensor data and motion capture data was tested using the Shapiro–Wilk test (α = 0.01). In addition, to check for the presence of heteroscedasticity in the differences, Kendall’s τ was calculated from the differences. When Kendall’s τ was greater than 0.1, heteroscedasticity was assumed to be present [19]. To quantify the absolute difference, statistical indices in the Bland–Altman plots were calculated for the overall, walking, and running conditions. When the differences were non-normally distributed or when the heteroscedasticity of the differences was present, median, 95-percentile range, and percentage interquartile range (IQR%) of the differences were utilized as the statistical indices for quantifying absolute difference. The 95-percentile range was defined as a range from 2.5-percentile to 97.5-percentile. IQR% was defined as an interquartile range of the difference divided by the median of the two means. However, the IQR% of the sagittal angle_IC_ and frontal angle_IC_ could not be calculated because these gait parameters’ average values were normally distributed around 0, making them inappropriate to evaluate in percentages.

## 3. Results

The status of trial execution and data acquisition is shown in Appendix A. A priori power analysis suggested a required trial sample size of N = 117, with power = 0.8, alpha = 0.05, null assumption of ICC = 0.7, and an alternative hypothesis = 0.8, based on the formula provided by Zou [20]. We obtained 125 average gait-parameter values as samples for both feet from each trial. Thus, the sample size required by a priori power analysis was satisfied. Thus, the sample size required by a priori power analysis was satisfied. 

### 3.1. Gait-cycle Detection Ratio in ORPHE ANALYTICS

The number of gait cycles for the analysis that were detected by the motion sensor data in each trial is shown in Appendix A. As regards the plantar-embedded motion sensor, the gait-cycle frequencies detected by the motion sensor and optical motion capture data were 5931 and 5981, respectively, and the gait-cycle detection ratio was 99.16%. Regarding the instep-mounted motion sensor, the gait-cycle frequencies detected by the motion sensor and optical motion capture data were 6343 and 6372, respectively, and the gait-cycle detection ratio was 99.54% (Appendix A). 

### 3.2. Gait-cycle Percentages, including Outliers

The percentages of gait-cycle data excluded from the analysis as outliers were 10.84% and 9.52% for the plantar-embedded and instep-mounted motion sensors, respectively (Appendix A).

### 3.3. Relative Validity of ORPHE ANALYTICS against Optical Motion Capture

The gait-parameter ICCs from ORPHE ANALYTICS and optical motion capture were shown in Table 2. Stride length, stride duration, stride frequency, stride speed, vertical height, stance phase duration, swing phase duration, and sagittal angle_IC_ exhibited excellent relative validities (ICC > 0.9) in both plantar-embedded and instep-mounted motion sensors. The frontal angle_IC_ demonstrated moderate (ICC = 0.566–0.627), while the sagittal angle_TO_ exhibited good (ICC = 0.892–0.833). 

### 3.4. Absolute Difference between ORPHE ANALYTICS and Optical Motion Capture

The Bland–Altman plots and histograms of the gait parameters from ORPHE ANALYTICS and the optical motion capture were shown in Figure 2 and Figure 3. The differences in gait parameters from motion sensor data and motion capture data for all gait parameters were non-normally distributed (*p* < 0.01 at Shapiro–Wilk test) or showed heteroscedasticity (Kendall’s τ > 0.1). Thus, the statistical indices were calculated based on percentile and shown in Table 3. Stride duration and stride frequency exhibited a small IQR% (IQR% = 0.2). Stride length, stride speed (plantar-embedded), vertical height (plantar-embedded), stance phase duration, swing phase duration, and sagittal angle_TO_ demonstrated moderate (IQR% = 6.1–9.5). The IQR% of stride speed (instep-mounted) was 13.4. The vertical height’s IQR% was extremely larger for the instep-mounted model compared with that for the plantar-embedded model (Table 3). Proportional errors were visually recognized for stride length and speed (Figure 2b,d and Figure 3b,d).

## 4. Discussion

In this study, ORPHE ANALYTICS-derived gait parameters were compared with those from optical motion capture data during walking and running tasks at speeds of 2–12 km/h in nine healthy adults. The relative validities of the stride length, stride duration, stride frequency, stride speed, vertical height, stance phase duration, swing phase duration, and sagittal angle_IC_ for both plantar-embedded and instep-mounted motion sensors were excellent (ICC > 0.9) with respect to the optical motion capture (Table 2). However, those of the sagittal angle_TO_ and sagittal angle_IC_ were good, while those of the frontal angle_IC_ were moderate. These results partially support our hypothesis. 

The relative validities of ORPHE ANALYTICS-derived gait parameters were not low compared with those of a previous meta-analysis involving commercial products such as Physilog, Mobility Lab, and Stryd, among others (ICC: stance phase time, 0.81–0.97; swing phase time, 0.56–0.81; stride duration, 0.55–0.99; stride frequency, 0.96–0.99; stride length, 0.75–0.99) [21]. The Bland–Altman plots revealed that stride length and speed had proportional relationships, and as the values increased, those calculated using ORPHE ANALYTICS became smaller than those calculated using optical motion capture. The results of this study, based on treadmill use by healthy adults, provide the validity standard for the future use of ORPHE ANALYTICS.

### 4.1. Differences between Plantar-Embedded and Instep-Mounted Motion Sensors

Regarding the mounting position, ORPHE ANALYTICS’ validity relative to optical motion capture was generally better for the plantar-embedded sensor than for the instep-mounted motion sensor (Table 2 and Table 3). Major differences were observed primarily in vertical height, stride length, and stride speed. A previous study reported that the error against the reference was smaller for a sensor embedded in the shoe midsole than for that mounted on the shoe instep [12,22]. This was due to an error in the integral calculation caused by the relatively high vibration susceptibility of the sensor mounted on the shoe instep [12]. Although we used dedicated mounting equipment to fix the motion sensor to the shoelace, complete suppression of the influence of vibration was difficult. While dedicated mounting equipment is advantageous in that the sensor can be attached to a variety of shoe types, the original shoes with space for storing the sensor have the advantage of higher gait analysis validity.

### 4.2. Less Agreement in Sagittal Angle_TO_, Swing Phase Duration, and Stance Phase Duration

The Bland–Altman plots revealed a relatively large IQR% in swing phase duration during walking as well as stance phase duration and sagittal angle_TO_ during running (Table 3). These errors were caused by differences in toe-off timing detection. Toe-off timing detection accuracy directly influences these parameters. Due to walking’s short swing phase duration, the time deviation of the toe-off timing detection relative to the swing phase duration becomes larger. Hence, the IQR% of the swing phase duration during walking is more prominent, exhibiting a trend similar to that in previous studies [6,23]. Conversely, the IQR% of the swing phase duration was slightly smaller during running because the swing phase duration was relatively longer. Even in previous motion-sensor-based gait analysis, the rule-based detection of toe-off timing from motion sensor data has been reported to be complex and inaccurate [13,24]. Hamacher et al. [25] found that the standard deviation of the angular velocity of the foot during walking is smaller in the middle of the swing phase and is larger in the early swing phase (immediately after the toe-off). The large variation in the foot motion around the toe-off may be a factor that makes it difficult to detect the timing of the toe-off. The difficulty in detecting toe-off timing may also influence the large error of the sagittal angle_TO_ during running.

### 4.3. The Proportional Errors in Stride Speed and Stride Length

As the treadmill speed increased, the ORPHE ANALYTICS-derived stride speed and length tended to be smaller than those calculated using optical motion capture data (Figure 2b,d and Figure 3b,d). These results are consistent with those reported in previous studies that analyzed running using motion sensors [12,26]. Falbriard et al. [26] noted certain limitations in estimating stride speed through the simple integration of accelerations during running. The reason why the error in the integration process increases at high speeds is thought to be due to the increase in signal, which has a relatively higher frequency than the sampling frequency of the sensor data. This argument is supported by reports that the smaller the sampling frequency of the sensor data, the larger the stride length error [27]. Considering these previous studies, it is reasonable to assume that the trend observed in this study is for errors to increase as speed increases. Falbriard et al. [26] attempted a correction using a linear function for the stride speed estimated by integrating 500-Hz sampling frequency accelerations to reduce the error associated with the speed. Considering these arguments, we suggest the following caution when interpreting the values: first, high relative validity implies its suitability for explaining changes in relative values, for example, evaluating changes in the same person’s speed. However, to refer to absolute values, a better estimation can be made by correcting for proportional agreement. When stride speed and length were corrected based on the regression lines obtained from this study’s results, the IQR% decreased to less than 5% (Appendix A). Interpreting stride speed and length using ORPHE ANALYTICS based on the assumption of existent proportional differences is necessary, especially during running.

### 4.4. Limitations

This study had certain limitations. ORPHE ANALYTICS validation was conducted using a treadmill because a conventional optical motion capture was used for comparison, and obtaining many samples for walking and running was necessary. In many clinical situations, treadmill-based gait is used as a task and measured using motion capture to measure speed-controlled data. In our validation, we initially provided a standard for the comparison of ORPHE ANALYTICS with optical motion capture data during treadmill-based walking and running.

Only healthy young adults participated in this study. To examine a wide range of gait speeds and foot-strike patterns, we selected healthy young adult participants with a high potential for exercise. However, gait patterns vary with age and disease status. Further validation is necessary for older people and patients with diseases whose gait patterns may affect gait-parameter calculation using motion sensors [6]. 

## 5. Conclusions

In this study, we evaluated the relative validity of gait parameters calculated using the motion-sensor-based gait analysis system ORPHE ANALYTICS against those calculated using optical motion capture in a 2–12 km/h gait involving nine healthy young adults.

Stride duration, stride length, stride frequency, stride speed (plantar-embedded), vertical height (plantar-embedded), stance phase duration, swing phase duration, and sagittal angle_IC_ exhibited excellent relative validity. However, sagittal angle_TO_ and frontal angle_IC_ exhibited good and moderate relative validity, respectively.

ORPHE ANALYTICS enables gait analysis regardless of the measurement environment and is expected to be applied in daily-life measurements. This study’s results will serve as a validity standard for the future use of this gait analysis system.

## Figures and Tables

**Figure 1 sensors-23-00331-f001:**
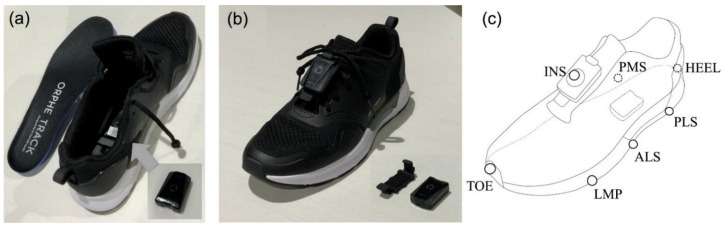
(**a**) Motion sensor embedded in the shoe midsole; (**b**) Motion sensor mounted on the shoe instep with dedicated mounting equipment; (**c**) Marker position on the shoe. (**a**) The shoe has space for sensor placement, and the center of the sensor is located 40% of the shoe’s length from the heel edge. (**b**) A motion sensor mounted on the shoe instep using dedicated mounting equipment that can be fixed to the shoelace. (**c**) The names and position definitions of the reflective markers are as follows: LMP, the lateral edge of the metatarsophalangeal joint; TOE, the most anterior edge of the shoe; PLS, the posterior lateral side of motion sensor; ALS, the anterior lateral side of motion sensor; PMS, the posterior medial side of motion sensor; HEEL, the most posterior edge of the shoe midsole; INS, the center of the sensor mounted on the instep equipment of the shoe.

**Figure 2 sensors-23-00331-f002:**
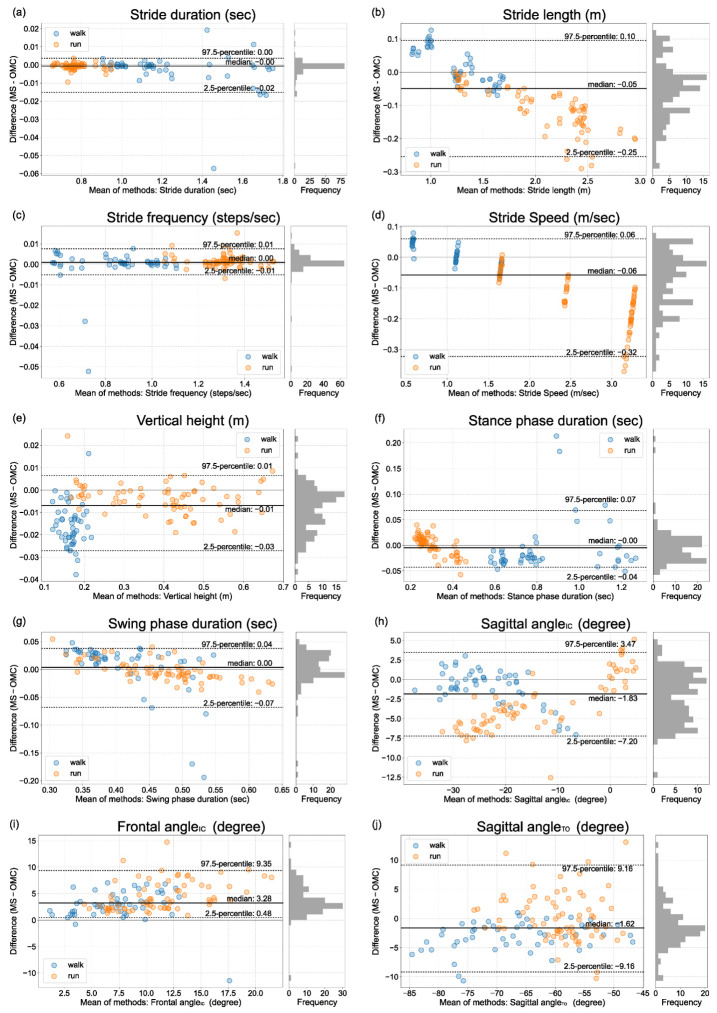
Bland–Altman plot of gait parameters from the motion sensor embedded in the shoe midsole (plantar-embedded) and optical motion capture. (**a**) stride duration, (**b**) stride length, (**c**) stride frequency, (**d**) stride speed, (**e**) vertical height, (**f**) stance phase duration, (**g**) swing phase duration, (**h**) sagittal angle_IC_, (**i**) frontal angle_IC_, (**j**) sagittal angle_TO_, MS: motion sensor, OMC: optical motion capture.

**Figure 3 sensors-23-00331-f003:**
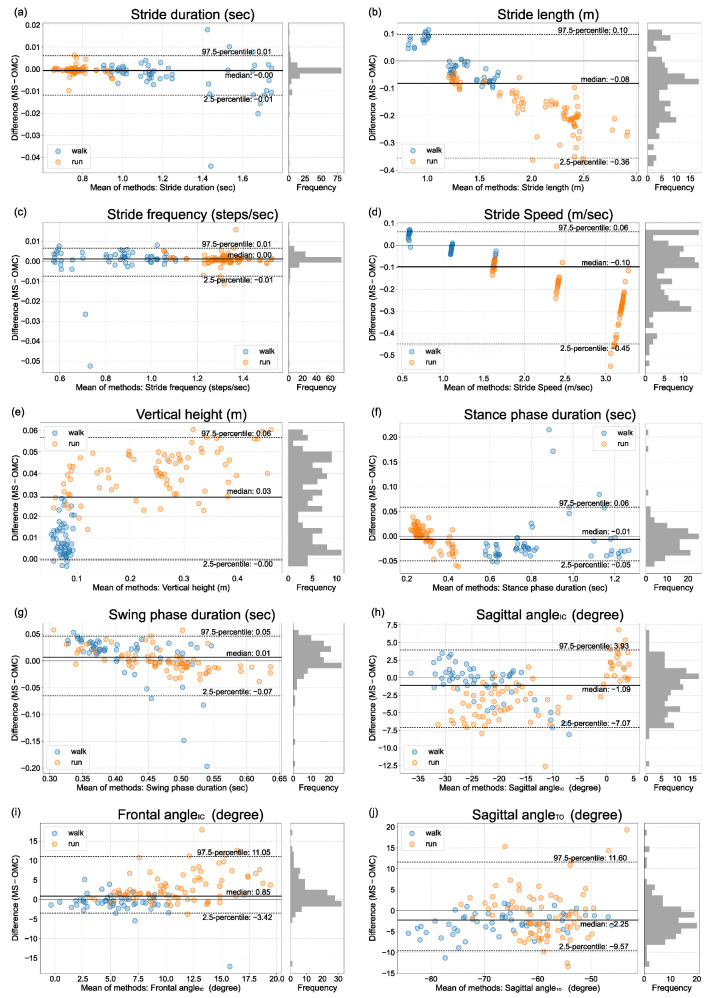
Bland–Altman plot of gait parameters from the motion sensor mounted on the shoe instep (instep-mounted) and optical motion capture (**a**) stride duration, (**b**) stride length, (**c**) stride frequency, (**d**) stride speed, (**e**) vertical height, (**f**) stance phase duration, (**g**) swing phase duration, (**h**) sagittal angle_IC_, (**i**) frontal angle_IC_, (**j**) sagittal angle_TO_, MS: motion sensor, OMC: optical motion capture.

**Table 1 sensors-23-00331-t001:** Gait parameters and their definitions.

Parameter Name	Unit	Explanation	Definition of Motion Capture Data Analysis	Name in ORPHE ANALYTICS *
Stride duration	sec	A time required for one gait cycle	A time required for IC to next IC	Stride duration
Stride length	m	Anteroposterior displacement of the foot during one gait cycle	Difference between the anteroposterior position of the motion sensor center at IC and next IC + (stride duration) × (treadmill speed)	Stride length
Stride frequency	steps/ sec	Number of gait cycles per 1 second	The inverse of stride duration	Stride frequency
Stride speed	m/sec	Average speed of the foot during one gait cycle	Stride length/stride duration	Stride speed
Vertical height	m	Maximum height of foot during one gait cycle	Maximum height of the center of motion sensor during one gait cycle	Vertical height
Stance phase duration	sec	A time that the foot is in contact with the ground during one gait cycle	Time required for IC to next TO	Ground contact time
Swing phase duration	sec	A time that the foot is in the air during one gait cycle	Time required for TO to next IC	Swing time
Sagittal angle_IC_	degree	The angle between ground and foot in the sagittal plane at IC	The angle between the motion sensor’s anteroposterior axis and its projection vector to the horizontal plane at IC	Strike angle
Frontal angle_IC_	degree	The angle between ground and foot in the frontal plane at IC	The angle between the motion sensor’s mediolateral axis and its projection vector to the horizontal plane at IC	Pronation
Sagittal angle_TO_	degree	The angle between ground and foot in the sagittal plane at TO	The angle between the motion sensor’s anteroposterior axis and its projection vector to the horizontal plane at TO	Toe off angle

Note: IC: initial foot contact, TO: toe-off, *: cited at https://orphe.io/en/analytics accessed on 1 October 2022.

**Table 2 sensors-23-00331-t002:** Intraclass correlation coefficients of the gait parameters from ORPHE ANALYTICS and optical motion capture.

Condition	Overall				Walking				Running			
Position	Plantar-Embedded		Instep-Mounted		Plantar-Embedded		Instep-Mounted		Plantar-Embedded		Instep-Mounted	
Parameter	ICC_(2,1)_	95%CI	ICC_(2,1)_	95%CI	ICC_(2,1)_	95%CI	ICC_(2,1)_	95%CI	ICC_(2,1)_	95%CI	ICC_(2,1)_	95%CI
Stride length	0.983	(0.949, 0.992)	0.963	(0.794, 0.986)	0.980	(0.964, 0.989)	0.971	(0.950, 0.983)	0.927	(0.000, 0.984)	0.855	(−0.022, 0.968)
Stride duration	1.000	(1.000, 1.000)	1.000	(1.000, 1.000)	0.999	(0.999, 1.000)	1.000	(0.999, 1.000)	1.000	(0.999, 1.000)	1.000	(1.000, 1.000)
Stride frequency	1.000	(1.000, 1.000)	1.000	(1.000, 1.000)	0.999	(0.998, 0.999)	0.999	(0.998, 0.999)	1.000	(0.998, 1.000)	1.000	(0.999, 1.000)
Stride speed	0.991	(0.962, 0.996)	0.979	(0.848, 0.993)	0.996	(0.993, 0.998)	0.993	(0.987, 0.996)	0.932	(0.001, 0.986)	0.855	(−0.022, 0.969)
Vertical height	0.996	(0.979, 0.999)	0.963	(0.376, 0.990)	0.750	(−0.060, 0.926)	0.572	(−0.019, 0.816)	0.998	(0.991, 0.999)	0.885	(−0.014, 0.976)
Stance phase duration	0.993	(0.991, 0.995)	0.993	(0.991, 0.995)	0.974	(0.955, 0.985)	0.975	(0.958, 0.986)	0.920	(0.792, 0.965)	0.917	(0.834, 0.959)
Swing phase duration	0.907	(0.870, 0.934)	0.904	(0.867, 0.931)	0.766	(0.625, 0.859)	0.764	(0.626, 0.856)	0.976	(0.924, 0.990)	0.976	(0.949, 0.988)
Sagittal angle_IC_	0.939	(0.820, 0.972)	0.945	(0.895, 0.968)	0.947	(0.896, 0.971)	0.926	(0.874, 0.957)	0.897	(0.422, 0.967)	0.913	(0.676, 0.967)
Frontal angle_IC_	0.566	(−0.064, 0.817)	0.627	(0.474, 0.736)	0.541	(0.010, 0.786)	0.607	(0.407, 0.751)	0.475	(−0.098, 0.801)	0.507	(0.044, 0.761)
Sagittal angle_TO_	0.892	(0.843, 0.925)	0.833	(0.752, 0.885)	0.922	(0.333, 0.977)	0.921	(0.439, 0.975)	0.715	(0.480, 0.854)	0.577	(0.294, 0.768)

Note: 95% CI: 95% confidence interval, ICC_(2,1)_ values were interpreted as follows: excellent (>0.90), good (0.75–0.90), moderate (0.50–0.75), and poor (<0.50) [17].

**Table 3 sensors-23-00331-t003:** Statistical indices of the Bland–Altman plots.

Condition	Overall				Walking				Running			
Position	Plantar-Embedded		Instep-Mounted		Plantar- Embedded		Instep-Mounted		Plantar-Embedded		Instep-Mounted	
Parameter	Median [95-percentile Range]	IQR%	Median [95-percentile Range]	IQR%	Median [95-Percentile Range]	IQR%	Median [95-Percentile Range]	IQR%	Median [95-Percentile Range]	IQR%	Median [95-Percentile Range]	IQR%
Stride length (m)	−0.049 [−0.254, 0.095]	6.3	−0.082 [−0.356, 0.097]	9.5	−0.003 [−0.057, 0.107]	6.9	−0.018 [−0.093, 0.108]	9.6	−0.110 [−0.266, −0.007]	4.5	−0.170 [−0.364, −0.055]	4.9
Stride duration (s)	−0.001 [−0.015, 0.004]	0.2	−0.001 [−0.012, 0.006]	0.2	−0.001 [−0.017, 0.009]	0.3	−0.001 [−0.019, 0.009]	0.4	−0.001 [−0.005, 0.002]	0.1	−0.001 [−0.004, 0.005]	0.1
Stride frequency (steps/s)	0.001 [−0.005, 0.008]	0.2	0.001 [−0.007, 0.006]	0.2	0.001 [−0.022, 0.007]	0.3	0.002 [−0.019, 0.007]	0.4	0.001 [−0.003, 0.009]	0.2	0.001 [−0.007, 0.005]	0.1
Stride speed (m/s)	−0.057 [−0.323, 0.060]	8.3	−0.098 [−0.450, 0.061]	13.4	−0.001 [−0.059, 0.061]	5.5	−0.012 [−0.092, 0.066]	8.1	−0.143 [−0.335, −0.006]	4.2	−0.235 [−0.464, −0.069]	5.0
Vertical height (m)	−0.007 [−0.027, 0.006]	6.1	0.029 [−0.001, 0.057]	32.4	−0.017 [−0.031, −0.001]	6.2	0.007 [−0.002, 0.027]	13.8	−0.003 [−0.016, 0.007]	1.7	0.041 [ 0.022, 0.060]	6.1
Stance phase duration (m)	−0.005 [−0.043, 0.068]	8.8	−0.007 [−0.050, 0.058]	9.5	−0.024 [−0.046, 0.157]	2.6	−0.025 [−0.048, 0.143]	2.8	0.006 [−0.034, 0.031]	7.2	0.003 [−0.048, 0.030]	8.0
Swing phase duration	0.004 [−0.068, 0.038]	7.6	0.007 [−0.065, 0.046]	7.6	0.021 [−0.147, 0.037]	4.8	0.022 [−0.127, 0.044]	4.7	−0.006 [−0.033, 0.034]	4.1	−0.004 [−0.033, 0.047]	4.4
Sagittal angle_IC_ (degree)	−1.833 [−7.204, 3.469]	N/A	−1.088 [−7.075, 3.929]	N/A	−0.432 [−6.438, 2.171]	N/A	−0.085 [−6.894, 3.211]	N/A	−4.091 [−7.551, 3.830]	N/A	−2.276 [−7.142, 4.586]	N/A
Frontal angle_IC_ (degree)	3.281 [0.485, 9.351]	N/A	0.854 [−3.425, 11.050]	N/A	2.942 [−0.559, 6.801]	N/A	−0.668 [−4.775, 3.508]	N/A	3.622 [1.328, 9.903]	N/A	1.945 [−1.220, 11.428]	N/A
Sagittal angle_TO_ (degree)	−1.616 [−9.160, 9.164]	−6.3	−2.255 [−9.573, 11.604]	−8.8	−2.825 [−9.388, 0.424]	−4.6	−2.739 [−8.420, 1.328]	−5.1	−0.294 [−7.589, 10.043]	−9.3	−1.227 [−10.035, 14.424]	−12.8

Note: N/A: not available (the IQR% of sagittal angle_IC_ and frontal angle_IC_ were not calculated because those values were distributed positively and negatively and their averages were close to 0, making them inappropriate to evaluate in percentages).

## Data Availability

The data analyzed in this manuscript will be made available from the corresponding author upon reasonable request.

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
