# Peer review of "Validity of Spatio-Temporal Gait Parameters in Healthy Young Adults Using a Motion-Sensor-Based Gait Analysis System (ORPHE ANALYTICS) during Walking and Running"

_sensors, 2022, doi:10.3390/s23010331_

Round 1

Reviewer 1 Report

Title: I would change the title to say what you measured (tempo-spatial)  and tested (treadmill gait), for example; ”Validity of the ORPHE ANALYTICS motion sensor system in quantifying healthy young adults tempo-spatial parameters during treadmill walking and running”

Line 76: you wrote “power analysis showed you need sample of 117 and you sampled only 9 participants”. Can you please explain. Or do you mean 117 gait cycles. Please explain in text.

Line 126: please provide reference to the smoothing cut of frequency of 10Hz.

Line 139: please indicate how you measured the height of the treadmill surface?

Statistical analysis: please indicate if you checked for normal distribution and which test you used. please not that you need to check for the presence of heteroscedasticity in the data (you can use the Kendall rank correlation coefficient τ or visual inspection of the B&A plot). When τ > 0.1 a logarithmic transformation of the data needs to be performed before calculating bias and 95% upper and lower limits of agreement.

Statistical analysis: in line 155 you said the first 15 gait cycles were discarded. Please indicate how many gait cycles you used in your analysis for each participant and speed

Line 252: you wrote “This 251 study’s results, based on treadmill use by healthy adults, endorse the reliability of ORPHE 252 ANALYTICS in clinical applications.”, please revise as you cannot say reliability for clinical application. You examined validity. To know the reliability for clinical setting to be used in interventions you need test-retest reliability design and measure the minimum detectable change (MDC). 

Line 267: Absolute reliability is the degree to which repeated measurements vary for individuals. I am not convinced that you did this. To me you just compared the outcome of two different systems i.e validity. Absolute reliability is assessed by calculating standard error of measurement (SEM), which I can’t see in your results.

Discussion: paragraph 281-296: can you please suggest a cut-off gait velocity that above that velocity the ORPHE system may be less useful in clinical setting. Need some more critical thinking here on why at high velocities the two systems agree less, it looks as it relates to stride length and not stride duration.  Also please explain/suggest what is the clinical application here.

Reviewer 2 Report

1. Authors should provide a block diagram of the proposed approach.

2. The experiment was done on less number of subjects. In future, try to enhance this work.

3. before starting this kind of work, you should follow ethical practice, like is there any approval taken from the ethical committee for data taken from human or animal.

4. Authors should read cite the related work

Bijalwan, V., Semwal, V. B., & Mandal, T. K. (2021). Fusion of multi-sensor-based biomechanical gait analysis using vision and wearable sensor. IEEE Sensors Journal21(13), 14213-14220.

Reviewer 3 Report

The purpose of this study was to examine the validity of the gait parameters calculated using ORPHE ANALYTICS relative to those calculated using conventional optical motion capture.
The results showed that 8 items (stride duration, stride length, stride frequency, stride speed, vertical height, stance phase duration, swing phase duration, and sagittal angleIC) exhibited excellent relative validities, whereas sagittal angleTO and frontal angleIC demonstrated good and moderate relative validity, respectively. Thus, the authors conclude that that the sensors are feasible for gait analysis outside the laboratory setting. Although this study could be of interest for Sensors readership, the manuscript should be revised.

General comments
1) Nine healthy volunteers participated in the study. However, an a priori power analysis suggested a required trial sample size of N=117 (with power=0.8, alpha=0.05, null assumption of ICC=0.7, and alternative hypothesis=0.8). Up to now I was of the opinion that the sample size calculation indicates the size of the group, i.e. participants. If that is the case, the number of nine participants would be too small. Furthermore, it is not clear which test family was used. Please explain.

2) Hamacher et al. (2017) found signs of phase-dependent functional variability particularly in the swing phase of gait (i.e. reduced variability in the time-continuous foot kinematics in mid swing during normal walking where also the minimum toe clearance event occurs). Please discus your results in light of this observation.
•    Hamacher, D., Hamacher, D., Müller, R., Schega, L., & Zech, A. (2017). Exploring phase dependent functional gait variability. Human movement science, 52, 191-196.

Specific comments
•    Line 80-82: Please delete “Interventionary studies…”
•    Line 111: Please specify “UY and NY”.
•    Legend Figure 2: Please use uniform font sizes.

Round 2

Reviewer 1 Report

I am still confused regarding the following: Lines 203–207: "A priori power analysis suggested a required trial sample size of N=117, with power=0.8,......."

I guess you define a trial as a speed condition i.e 2kph, 4kph, 6kph etc. This is a bit confusing as in the protocol you call it a task. I suggest to use one definition, and maybe in the protocol say a walking speed was considered a trial, and in the analysis you can say that for each walking/running trial a power analysis showed the need of 117 gait cycles. Is this correct?

Reviewer 3 Report

Thank you for addressing all my comments. I have only one minor comment:

Line 126, 210, 211: "iswas"? Please correct.
